# Beta-Blockers of Different Generations: Features of Influence on the Disturbances of Myocardial Energy Metabolism in Doxorubicin-Induced Chronic Heart Failure in Rats

**DOI:** 10.3390/biomedicines12091957

**Published:** 2024-08-28

**Authors:** Igor Belenichev, Olexiy Goncharov, Nina Bukhtiyarova, Oleh Kuchkovskyi, Victor Ryzhenko, Lyudmyla Makyeyeva, Valentyn Oksenych, Oleksandr Kamyshnyi

**Affiliations:** 1Department of Pharmacology and Medical Formulation with Course of Normal Physiology, Zaporizhzhia State Medical and Pharmaceutical University, 69035 Zaporizhzhia, Ukraineolegk181@gmail.com (O.K.); 2Department of Clinical Laboratory Diagnostics, Zaporizhzhia State Medical and Pharmaceutical University, 69035 Zaporizhzhia, Ukraine; 3Department of Medical and Pharmaceutical Informatics and Advanced Technologies, Zaporizhzhia State Medical and Pharmaceutical University, 69035 Zaporizhzhia, Ukraine; 4Department of Histology, Cytology and Embryology, Zaporizhzhia State Medical and Pharmaceutical University, 69035 Zaporizhzhia, Ukraine; 5Broegelmann Research Laboratory, Department of Clinical Science, University of Bergen, 5020 Bergen, Norway; 6Department of Microbiology, Virology, and Immunology, I. Horbachevsky Ternopil National Medical University, 46001 Ternopil, Ukraine

**Keywords:** chronic heart failure, beta-blockers, energy metabolism, mitochondria, Hypertril

## Abstract

Beta-blockers are first-line drugs in the treatment of chronic heart failure (CHF). However, there is no consensus on the specific effects of the beta-blockers of the I-III generation on energy metabolism in CHF. The aim of this study is to conduct a study of beta-blockers of different generations on myocardial energy metabolism in experimental CHF. CHF was modeled in white outbred rats by administering doxorubicin. The study drugs were administered intragastrically—new drug Hypertril (1-(β-phenylethyl)-4-amino-1,2,4-triazolium bromide)-3.5 mg/kg, Metoprolol—15 mg/kg, Nebivolol −10 mg/kg, Carvedilol 50 mg/kg, and Bisoprolol, 10 mg/kg. In the myocardium, the main indices of energy metabolism were determined—ATP, ADP, AMP, malate, lactate, pyruvate, succinate dehydrogenase (SDH) activity, and NAD-dependent malate dehydrogenase (NAD-MDH) activity. Traditional second-generation beta-blockers (Metoprolol and Bisoprolol) did not affect the studied indices of energy metabolism, and third-generation beta-blockers with additional properties—Carvedilol and, especially, Nebivalol and Hypertril—improved myocardial energy metabolism. The obtained results will help to expand our understanding of the effect of beta-blockers of various generations used to treat cardiovascular diseases on energy metabolism, and are also an experimental justification for the practical choice of these drugs in the complex therapy of CHF.

## 1. Introduction

It is known that, in recent years, the main cause of mortality and disability in the population of industrially developed countries is cardiovascular diseases [1,2]. Among all cardiovascular diseases, chronic heart failure (CHF) is a complex and major health problem in many countries. Despite the progress achieved over the past 20 years in the treatment of CHF, the problem remains relevant [3,4,5]. CHF is becoming one of the leading problems of modern medicine and is characterized by an extremely unfavorable prognosis. Thus, the annual mortality rate among patients with functional class III-IV CHF reaches 60%, and only half of less severe patients survive for 5 years from the date of diagnosis [6,7,8,9]. To date, recommendations have been developed for the treatment of CHF, which include the prescription of angiotensin-converting enzyme (ACE) inhibitors, diuretics, cardiac glycosides, and beta-blockers [10,11]. Beta-blockers, along with ACE inhibitors, are considered first-line drugs in the treatment of CHF, as they can improve the survival rates and hospitalization of patients, effectively increase the ejection fraction (EF), and reduce the mass and sphericity of the left ventricle (LV) [12]. Beta-blockers have the greatest effect on reducing heart rate, myocardial contractility, and cardiomyocyte O_2_ requirement, which provides them with cardioprotective properties [13]. Currently, depending on the pharmacological action, there are three generations of beta-blockers.

First-generation beta-blockers—non-selective block β1- and β2-receptors (Propranolol (Anaprilin), Nadolol, Timolol)—exhibit negative effects on carbohydrate and lipid metabolism, the central nervous system, and erectile function in men, as well as increase the tone of bronchial smooth muscles, vascular walls, and myometrium, which significantly limits their use in clinical practice. Second-generation β-blockers are cardioselective since they block only β1-receptors (Metoprolol, Bisoprolol, Atenolol), which have fewer side effects compared to first-generation drugs and have more favorable tolerability with long-term use and a convincing evidence base for long-term prognosis life during the treatment of CHF. Third-generation β-blockers, which can be either highly selective for β1 receptors (Nebivalol, Betaxolol) or non-selective (Carvedilol, Labetalol, Carteolol), and have anti-ischemic, endotheliotropic, antioxidant, antiproliferative, antihypertrophic, and antiapoptotic activity, with minimal side effects. Also, first- and second-generation beta-blockers are called traditional, and third-generation beta-blockers have additional pharmacological properties [14,15,16].

A proposed medication, a result of our multi-year efforts—Hypertril (1-(β-phenylethyl)-4-amino-1,2,4-triazolium bromide)—belongs to class IV toxicity (LD50 is 683.4 mg/kg after intragastric administration to rats) and exhibits β1-blocking, vasodilatory, antihypertensive, and cardioprotective properties [17]. The synthesis and standardization of the substance “Hypertril”, as well as the technology of parenteral solutions, have been developed by SPA “Pharmatron” in collaboration with the scientific and technological complex “Institute of Single Crystals” of the National Academy of Sciences of Ukraine (certificate No. 2, series 020213). The State Expert Center of the Ukrainian Ministry of Health has decided that phase 1 of Hypertril’s antihypertensive medication clinical trials has been successfully finished, and phase 2 is currently in progress. The first experimentally verified benefits of Hypertril for CHF have been reported, along with its advantages over other beta-blockers [18,19].

It is known that CHF is accompanied by mitochondrial dysfunction and persistent disturbances in myocardial energy metabolism leading to ATP deficiency, oxidative stress, and apoptosis [20,21]. Myocardial metabolic disorders can be considered as a link in the pathogenesis of CHF, as well as a factor in the progression of CHF. The cardiac effects of traditional beta-blockers and, especially, beta-blockers with additional properties suggest the possibility of the drugs influencing the energy metabolism of the myocardium. However, the results of studies are few and demonstrate an ambiguous effect of beta-blockers on energy metabolism depending on the class of the drug. Thus, Propranolol (Anaprilin), a non-selective beta-blocker, leads to disturbances in the energy metabolism of the myocardium due to the negative effect on mitochondria (inhibition of the II complex of the respiratory chain, collapse of the membrane potential of mitochondria) [22,23]. At the same time, Timolol had a mitoprotective effect and improved metabolism, which is directly related to the activity of myocardial mitochondria [24]. Data were obtained on the effect of a beta-blocker with additional (antioxidant) properties—Carvedilol—on energy metabolism, in particular, its effect on mitochondrial enzyme complexes and reactive oxygen species (ROS) production [25].

Based on the above, we can outline the main direction of the ongoing research, which, in our opinion, will allow us to form a theoretical foundation for a clearer understanding of the possible and potential mechanisms of action of beta-blockers of various generations on energy metabolism, which will, perhaps, resolve some practical issues associated with the treatment of CHF. Therefore, the purpose of theisstudy was to investigate the effect of beta-blockers of various generations (Bisoprolol, Metoprolol, Carvedilol, Nebivalol, and the new drug Hypertril) on the indicators of myocardial energy metabolism in a model of doxorubicin-induced CHF.

## 2. Materials and Methods

### 2.1. Animals

A total of 130 white outbred rats, weighing between 190 and 220 g, were used in the studies. They were taken from the vivarium of the Institute of Physiology and the Institute of Pharmacology and Toxicology of the Academy of Medical Sciences of Ukraine. A.A. Bogomolets were from the Ukrainian Academy of Medical Sciences. All animals were placed under a 14-day quarantine (acclimatization period). Every animal under quarantine had daily examinations for behavior and general health, and they were observed in their cages twice a day for morbidity and mortality. Animals that satisfied the experiment’s inclusion requirements were split into groups using the randomization technique prior to the study’s commencement. The animals were watched during the experiment, their appearance was described, and their deaths were noted. All manipulations were performed in compliance with the “European Applicable Protection of Vertebrate Animals used for Experimental and Scientific Purposes” and the regulations on the gathering of animals for biomedical investigations (Strasbourg, France, 1986, as revised in 1998). The ZSMU Commission on Bioethics decided to adopt the experimental study protocols and outcomes (Protocol No. 3, dated 22 March 2021).

### 2.2. Experimental Model

Chronic cardiac failure was replicated using a doxorubicin model [26]. The doxorubicin pharmacological model of chronic heart failure (CHF) is thought to be the most successful, causing most animals to develop severe and progressive CHF. When rats are given doxorubicin intraperitoneally at a cumulative dose of 15 mg/kg, given in 6 injections over a 14-day period, their left myocardium contracts less, its eccentric remodeling decreases, and progressive chronic heart failure develops. Figure 1 describes doxorubicin-induced cardiotoxicity mechanisms.

### 2.3. Drugs and Pharmacological Agents

Doxorubicin “Ebeve” 50 mg/25 mL (EBEWE Pharma Ges.mbH Nfg. KG, Unterach am Attersee, Austria) was utilized in this study. For 30 days following a 14-day administration of doxorubicin—Hypertril—at an experimentally supported dose of 3.5 mg/kg, all preparations were given intragastrically once daily as a suspension of 1% starch mucus [17]. Bisoprolol 10 mg/kg [27], Carvedilol 50 mg/kg [28], Nebivalol 10 mg/kg [29], and Metoprolol succinate 15 mg/kg [26,30]. Ten animals made up the intact group, while twenty animals each made up the control and experimental groups. The following drugs were used in this research: Carvedolol tablets (Salutas Pharma GmbH, Barleben, Germany), Bisoprolol tablets (Teva Pharmaceutical Industries, Ltd., Tel Aviv, Israel), Nebivalol tablets (AstraZeneca UK Ltd., Mölndal, Sweden), and Hypertril substance (Scientific and technological complex “Institute of Single Crystals” of the National Academy of Sciences of Ukraine, certificate N°2, series 020213).

### 2.4. Anesthesia

Rats from all experimental groups were removed from the study along with provided thiopental anesthesia (40 mg/kg). After that, blood samples from the celiac artery were taken for further examination.

### 2.5. Preparation of Biological Material Experimental Model

In a 1:10 ratio, cold 0.15 M KCl (+4 °C) was used to clean the heart. Once the extra fat and connective tissue were removed, along with the internal cavities’ blood arteries and clots, the heart was again cleaned in a 1:10 ratio using 0.15 M KCl (+4 °C). After that, it was ground up in liquid nitrogen until a powdery texture was obtained. One hundred milligrams of cardiac tissue that had been previously pulverized into a fine powder using liquid nitrogen was precisely weighed on a WT500 torsion balance (made in Moscow, Russia). The tissue powder was then well combined with 10.0 mL of a medium maintained at +2 °C. The concentration of these components in millimoles per liter (mmol/L) was 250 mmol/L of sucrose, 20 mmol/L of Tris-HCl buffer, and 1 mmol/L of EDTA, all of which were adjusted to a pH of 7.4. After that, the homogenate was put through a pre-centrifugation process in which big cell fragments were extracted using a Sigma 3–30 k refrigerated centrifuge (Osterode am Harz, Germany) set at +4 °C for 7 min at 1000 g. Using the same Sigma 3–30 k refrigerated centrifuge (Germany), the resultant supernatant was carefully collected and then put through a second centrifugation step at +4 °C for 20 min at 17,000× *g*. After completing this process, the supernatant was gathered and kept cold at −80 °C. For additional research, the thick mitochondrial precipitate that was formed during resuspension was used.

### 2.6. Biochemical Assays and Techniques

The concentration of ATP, ADP, AMP, malate, pyruvate, and lactate was used to assess the condition of cardiac energy metabolism, as were the rate at which the mitochondrial hole opens, the charge of the mitochondrial membrane, and the activity of succinate dehydrogenase and malate dehydrogenase.

#### 2.6.1. Determination of Adenyl Nucleotides by Thin-Layer Chromatography

The principle of the method is as follows. The method was based on the separation of ATP, ADP, and AMP in the dioxane–isopropanol–water–ammonia system on a thin layer of sorbent, followed by quantitative determination through direct spectrophotometry at 260 nm.

Reagents:-Isopropanol;-Dioxane;-Ammonia;-Plates for thin-layer chromatography on an aluminum substrate coated with a microfractionated silica gel sorbent.

Conducting research:

In total, 0.2 mL of protein-free tissue extract was applied to the starting line of the plate and chromatographed in the dioxane–isopropanol–water–ammonia system (4:2:4:1). ATP, ADP, and AMP were identified in ultraviolet light at UVC—365 nm. The samples were eluted with 4.0 mL of 0.1 N HCl and were then measured using a spectrophotometer at 260 nm. The content of ATP, ADP, and AMP (μmol/g tissue) was calculated from the calibration curve and adjusted per tissue sample.

#### 2.6.2. Quantitative Determination of Pyruvate Content Using the Zoch–Lamprecht Method

The principle of the method is as follows. In the presence of lactate dehydrogenase (LDH), pyruvate was reduced to lactate.
Pyruvate + NADH + H^+^ ↔ lactate + NAD^+^(1)

The amount of pyruvate used in the reaction was equimolar to the amount of NADH, and the decrease in NADH was determined at a wavelength of 340 nm.

Reagents:-0.5 M tris-HCl-aminomethane buffer, pH 7.6;-0.06 M NADH solution;-lactate dehydrogenase (activity 700 units/mg).

Conducting research:

In total, 0.8 mL of protein-free extract was added to 1.2 mL of Tris-HCl buffer. The reaction was started by adding 0.05 mL of LDH solution. The optical density was measured before the reaction was started (E1) and after 4 min (E2). The amount of pyruvate was calculated using the following formula:(2)$$C=\frac{\Delta E\cdot K\cdot V}{6.22}$$
where: ∆E (E2 − E1);

V—the final sample volume in the cuvette;K—the dilution factor of the sample relative to the tissue.

#### 2.6.3. Quantitative Determination of Malate Using the Hohorst Method

The principle of the method is as follows. In the presence of malate dehydrogenase (MDH), malate was converted to oxaloacetic acid. The binding of oxaloacetic acid with hydrazine–glycine buffer ensured the complete oxidation of malate.
Malate + NAD^+^ + hydrazine ↔ oxaloacetate-hydrazine + NADH + H_2_O(3)

The formation of the reduced form of NADH was equimolar to the amount of oxidized malate, and the increase in NADH was recorded at 340 nm.

Reagents:-0.4 M hydrazine–glycine buffer, pH 9.5;-0.05 M NAD+ solution;-malate dehydrogenase (activity 700 units/mg).

Conducting research.

In total, 0.2 mL of protein-free brain extract was placed in 2.2 mL of hydrazine–glycine buffer. The reaction was started by adding 0.2 mL of NAD^+^ solution. The optical density was measured before the reaction was started (E1) and after 4 min (E2). The amount of malate was calculated using the formula:(4)$$C=\frac{\Delta E\cdot K\cdot V}{6.22}$$
where: ∆E (E2 − E1);

V—the final sample volume in the cuvette;K—the dilution factor of the sample relative to the tissue.

#### 2.6.4. Determination of Succinate Dehydrogenase (SDH) Activity

The principle of the method is as follows.

Under the influence of SDH, potassium hexacyanoferroate (III) K_3_[Fe(CN)_6_], which is yellow, was reduced to colorless potassium hexacyanoferroate (II) K_4_[Fe(CN)_6_] by succinic acid. The enzyme activity was proportional to the amount of hexacyanoferroate (III) that was reduced.

Reagents:-0.1 M succinic acid;-25 mM potassium hexacyanoferroate (III);-150 mM sodium azide;-25 mM EDTA (pH 7.8);-0.1 M phosphate buffer (pH 7.8);-20% trichloroacetic acid (TCA).

Conducting research.

In total, 1 mL of 0.1 M phosphate buffer and 0.1 mL of solutions of succinic acid, EDTA, sodium azide, and distilled water were poured into centrifuge tubes. After this, 0.5 mL of the test tissue was added to the reaction medium and incubated for 5 min to inhibit cytochrome oxidase with sodium azide. The reaction was started by adding 0.1 mL of potassium hexacyanoferroate (III) solution. The samples were incubated for 10–15 min at 30 °C. After incubation, the reaction was stopped by immersing the samples in ice and adding 2 mL of 20% TCA. To control samples containing all components of the incubation mixture, TCA was added before the tissue homogenate was included. Thus, SDH in the control samples was completely denatured from the beginning of incubation, and no specific reduction occurred. After stopping the reaction and cooling, the samples were centrifuged at 2000 rpm (at 15 °C) for 15 min to precipitate the denatured protein. The supernatant was measured using a photometer at 420 nm. A mixture of 20% TCA and 0.1 M phosphate buffer (1:1) served as an optical control. To determine the content of potassium hexacyanoferroate (III) in the samples, a calibration curve was constructed based on the results of photometry of samples containing from 100 to 1000 μg of potassium hexacyanoferroate (III) in 4 mL of solution. The amount of potassium hexacyanoferroate (III) reduced during incubation was calculated from the difference in extinctions (Ek − Epr). Enzyme activity (nmol succinate/min per 1 mg protein) was calculated using the formula:A = 1000 · m/2M · a · t(5)
where: m is the amount of reduced potassium hexacyanoferroate (III) in the sample;

a—protein content in the sample, mg;M—molecular weight of potassium hexacyanoferroate (III);t—incubation time, min.

#### 2.6.5. Determination of Lactate Content Using the Hohorst Method

In the presence of lactate dehydrogenase (LDH), lactate was converted to pyruvate, and the binding of pyruvate formed during the reaction with hydrazine–glycine buffer promoted the complete oxidation of lactate.
Lactate + NAD^+^ + hydrazine → hydrazine-pyruvate + NADH + H_2_O(6)

The formation of the reducing form of NAD was equimolar to the amount of oxidized lactate, and the increase in NAD was recorded at 340 nm.

Reagents: -0.4 M hydrazine–glycine buffer pH 9.5;-0.05 M NAD solution;-lactate dehydrogenase (activity 700 units/mg).

Conducting research

In total, 0.2 mg of protein-free tissue extract was added to the incubation mixture, which consisted of 2.0 mL of hydrazine–glycine buffer and 0.2 mL of NAD solution. The optical density was measured before the reaction was started (E1) and after 4 min (E2). The amount of lactate was calculated using the formula:(7)$$C=\frac{\Delta E\cdot K\cdot V}{6.22}$$
where: ∆E (E2 − E1);

V—the final sample volume in the cuvette;K—the dilution factor of the sample relative to the tissue.

#### 2.6.6. Determination of NAD-Dependent Malate Dehydrogenase Activity

The principle of the method is as follows. The activity of NAD-dependent malate dehydrogenase was studied in the direct malate dehydrogenase reaction, where the amount of oxidized malate was equimolar to the amount of reduced NAD. The increase in NADH concentration in the samples was recorded at 340 nm. The reaction was carried out in an alkaline medium to promote a shift towards the oxidation of malate.

Reagents included incubation medium: sodium glycine buffer—85 mM (pH 10.0); D;L-sodium malate—85 mM; and NAD—25 mM.

The progress of determination is as follows. The reaction was started by adding 0.1 mL of tissue extract to 2.9 mL of incubation medium. The optical density was measured immediately (E1) and its increase was measured after 3 min (E2) at a wavelength of 340 nm. Enzyme activity was expressed in µmol of NADH formed per minute per 1 mg of protein (NAD/min per 1 mg of protein). Enzyme activity was calculated using the formula:X = EV/6.22a,(8)
where: E—change in optical density at 340 nm in 3 min;

V—final sample volume (3 mL);6.22 micromolar extinction coefficient of NADH at 340 nm;is the protein concentration in the sample.

#### 2.6.7. Opening of the Mitochondrial Pore (MP)

The determination procedure is as follows. A suspension of mitochondria (1 mg of protein per sample) was added to the incubation medium (70 mM sucrose, 5 mM HEPES, 70 mM KCl, 0.5–1 mM KH_2_PO_4_, pH 7.4). The opening of the mitochondrial membrane potential (MP) was determined at λ = 540 nm at 25 °C with constant stirring for 25 min.

The mitochondrial membrane potential (MPM) (Ψ) of mitochondrial charge was measured in the presence of safranin-O. The determination progress is as follows. The potential generated across the inner mitochondrial membrane was recorded on a spectrophotometer in dual-wave mode (511–533 nm), using safranin-O as a voltage-dependent probe (18 µM). Measurements were carried out in a 10 × 10 mm glass cuvette with a working volume of 2 mL. The measurements were conducted in a sodium environment, which contained 0.62 mM NaCl; 40 mM Caps (3-[cyclohexylamino]-1-propanesulfonic acid)-NaOH (pH = 10). The protonophore uncoupler FCCP (p-trifluoromethoxyphenylhydrazone) and the antiporter monensin were used to dissipate the potential. The swelling of mitochondria was recorded on a spectrophotometer as a decrease in the optical density of the mitochondrial suspension at 540 nm.

For biochemical studies, an Eppendorf BioSpectrometer (USA) and an Agilent Fluorescence Spectrophotometer (USA) were used.

### 2.7. Statistical Analysis

The program “STATISTICA^®^ for Windows” (StatSOFT, Hamburg, Germany) was used to perform statistical analyses. Group comparisons were evaluated using the Kruskal–Wallis criterion with subsequent Dunn adjustment, or one-way ANOVA or ANOVA for repeated measurements, followed by post hoc Bonferroni correction. A *p*-value of less than 0.05 was used to determine statistical significance.

## 3. Results

Biochemical studies revealed disturbances in myocardial energy metabolism and energy deficiency in the group of animals with CHF. Thus, a 14-day administration of doxorubicin led, on the 45th day of the experiment, to a decrease in the level of ATP in the myocardial cytosol by 50% and in mitochondria by 55%. In parallel, a decrease in the ADP content in the myocardial cytosol of rats with CHF by 42% was recorded against the background of an increase in the ADP level by 75.3% (Table 1).

Also, we recorded a 70% malate deficiency in the cytosolic fraction with a 62% decrease in the activity of mitochondrial NAD-dependent MDH (Table 2 and Table 3).

A course of administration of Hypertril tablets to rats with CHF resulted in a reduction in manifestations of secondary mitochondrial dysfunction. Thus, in animals treated with a course of Hypertril, there was a decrease in the opening of the mitochondrial pore (MP) by 51% (*p* < 0.05), as well as an increase in the charge of the inner membrane of the myocardial mitochondria by 82% (*p* < 0.05) compared to the group of untreated animals (Table 3). In these indicators, Hypertril is superior to the action of both traditional beta-blockers, Metoprolol and Bisoprolol, as well as Carvedilol and Nebivalol. Biochemical studies of the myocardium of rats with CHF made it possible to identify the features of the mitoprotective and anti-ischemic action of Hypertril. Thus, the ATP content in the cytosolic and mitochondrial fractions in rats treated with Hypertril increased (*p* < 0.05) by 42% and 33%, respectively. An increase of 24% in the ADP content and a decrease in AMP by 54% were observed in the cytosol of the heart (*p* < 0.05) compared with the corresponding indicators of the control group (Table 1, Table 2 and Table 3). In the cytosolic and mitochondrial fractions of the myocardial homogenate of animals with myocardial infarction under the influence of Hypertril, a decrease in lactate by 38.8 and 37%, respectively, was observed, which indicated a decrease in the activity of low-productive glycolysis (Table 2). At the same time, in the myocardial mitochondria of rats with CHF treated with Hypertril, the activity of SDH increased by 72% (*p* < 0.05) and NAD-MDH by 75.3% (*p* < 0.05) compared with the group of untreated animals. In the cytosolic fraction of the myocardial homogenate of rats with CHF, against the background of Hypertril administration, the levels of malate (by 90%) and pyruvate (by 24%) significantly increased. The positive changes in the myocardium of animals under the influence of Hypertril indicate a decrease in the manifestations of mitochondrial dysfunction and the activation of the compensatory cytosolic–mitochondrial shunts of ATP synthesis and a decrease in energy deficiency. The administration of Nebivalol to rats with CHF also had a positive effect on the energy metabolism of the myocardium. Thus, Nebivalol reduced mitochondrial swelling by 25% (*p* < 0.05) and increased the mitochondrial charge by 41% (*p* < 0.05) compared to the parameters of the group of untreated animals. Nebivalol increased the concentration of ATP in the cytosol and mitochondria of the myocardium of rats with CHF by 21% (*p* < 0.05) and 18% (*p* < 0.05), respectively, against the background of a decrease in AMP (*p* < 0.05) compared to the control. The administration of Nebivalol increased the activity of SDH by 44% (*p* < 0.05) and NAD-MDH by 21.5% (*p* < 0.05) in the myocardial mitochondria of rats with CHF compared to the control. The administration of Nebivalol resulted in a decrease in lactate (in the cytosol by 33% and in the mitochondria by 18%) (*p* < 0.05) and an increase in malate by 27% (*p* < 0.05) and pyruvate by 12% (*p* < 0.05) in the cardiac cytosol of rats with CHF compared to the group of untreated animals with CHF.

The administration of Carvedilol had a significant effect on the indices of mitochondrial swelling reduction (30%) and mitochondrial charge increase (39%) in the myocardium of rats with CHF (Table 3). Carvedilol also significantly increased the activity of SDH (15%) in the mitochondria of rats with CHF compared to the control. The administration of Carvedilol had no significant effect on the other studied indices of energy metabolism in the myocardium of rats with CHF (Table 1, Table 2 and Table 3). Traditional beta-blockers Metoprolol and Bisoprolol had no significant effect on the studied indices of energy metabolism in the myocardium of rats with CHF (Table 1, Table 2 and Table 3). The only thing that attracts attention in the group of rats with CHF that received a course of Bisoprolol in the mitochondrial fraction of the myocardium was a decrease in lactate by 14% (*p* < 0.05) compared to the control.

## 4. Discussion

Our results correspond to generally accepted ideas about disturbances in the energy supply of the myocardium under conditions of ischemia [31]. The obtained results on the change in the concentration of malate and the activity of NAD-MDH in the myocardium of rats with CHF indicate the possible inhibition of the malate–aspartate shuttle mechanism of the transport of reduced equivalents into the mitochondria [32,33] and the formation of secondary mitochondrial dysfunction. Among the causes of mitochondrial dysfunction in CHF are oxidative stress, the disruption of NO biosynthesis, the production of its cytotoxic derivatives, and the development of nitrosating stress [34,35,36]. It is currently known that the main manifestations of mitochondrial dysfunction are a decrease in the level of ATP in the cell, an increase in the level of lactate and a decrease in pyruvate, the activation of cell death mechanisms, and the production of reactive oxygen species (ROS) by mitochondria [37]. Currently, the effect of impaired ATP synthesis in mitochondria on the functional activity of the myocardium has been studied to the greatest extent [38]. It has been established that, with a decrease in the content of ATP in the mitochondria and cytosol of the myocardium by 10–20%, the activity of all energy-dependent processes decreases by 80%. The effects of an insufficient amount of ATP include the suppression of the disruption of ion pumps, ion homeostasis, and, accordingly, the contractile function of the heart [39]. The inhibition of energy production processes in the mitochondria of cardiomyocytes is accompanied by a weakening of lipid beta-oxidation, which results in a violation of lipid homeostasis in the cell and the accumulation of acyl-CoA thioesters, acylcarnitines, ceramides, and triglycerides, which enhance the formation of myocardial hypertrophy in CHF [40,41].

Under the influence of mitochondria-formed ROS, there is an increase in the opening of mitochondrial pores, expression, and release of proapoptotic proteins into the cytosol. The opening of the pores occurs due to the oxidation of the thiol groups of the cysteine-dependent region of the protein of the inner mitochondrial membrane (ATP/ADP antiporter) by cytotoxic derivatives of NO, which turns it into a permeable non-specific channel—a pore [42,43]. ROS generated by mitochondria also participate in the transmission of intracellular signals of receptors for endothelin, TGF-β1, PDGF, AT-II, FGF-2, etc. ROS are also capable of changing the activity of various transcription factors, including NF-κB, AP-1, and the proapoptotic protein p66Shc. In general, an increase in ROS production, by affecting the intracellular signaling mechanisms discussed above, can contribute to the activation of the inflammatory process in heart tissue and the development of hypertrophic and fibrotic changes [44,45].

Analyzing the obtained results of the biochemical studies of myocardial energy metabolism in experimental CHF and after the administration of Hypertril, it can be concluded that the starting mechanism of the anti-ischemic action of Hypertril is its effect on the dysfunction of the mitochondria of cardiomyocytes. Apparently, Hypertril, by reducing the damaging effect of ROS and free radicals on the SH-groups of the cysteine-dependent region of the protein of the inner mitochondrial membrane, prevents the opening of the mitochondrial pore and maintains the functional activity of the mitochondria, which subsequently improves the energy metabolism of the myocardium under ischemic conditions [42]. This statement is also confirmed by our previous study, which showed that Hypertril, unlike Metoprolol, Bisoprolol, Carvedilol, and Nebivalol, leads to a decrease in systolic and diastolic dysfunction, the restoration of autonomic mechanisms of heart rhythm regulation, and a decrease in the amplitude of the ST interval (*p* < 0.05), which, in combination with the restoration of the amplitude of the R wave, indicates the preservation of a high performance of cardiomyocytes in doxorubicin-induced CHF [19]. The mechanism of such an effect of Hypertril on energy metabolism parameters in rats with CHF is apparently associated not only with its β1-adrenergic blocking effect, but is also possibly realized through additional mechanisms identified earlier—antioxidant and NO-mimetic [46]. The viability of such assumptions is based on various studies that have shown that patients with mitochondrial disorders have a deficiency of NO, and the administration of NO-mimetics leads to an improvement in mitochondrial function and energy metabolism [47,48].

Carvedilol, a β-adrenergic receptor antagonist with strong antioxidant activity, provides a high degree of cardioprotection in various experimental models of ischemic heart injury. Data on the effect of Carvedilol on mitochondrial bioenergetic functions and ROS formation have been obtained. Thus, Carvedilol is able to reduce the formation of H_2_O_2_, increase the level of reduced glutathione, and restore mitochondrial respiration due to its antioxidant effect [49]. Carvedilol exhibits the properties of an ROS scavenger and also inhibits the formation of ROS in mitochondria due to “soft uncoupling” and a slight decrease in the potential of the mitochondrial membrane; it is able to directly protect the ultrastructure of mitochondria and reduce the Ca^2+^ overload of mitochondria, but does not affect the indicators of mitochondrial respiration after the 7-week administration of doxorubicin [50,51]. It has been shown that the direct mitoprotective properties of Carvedilol are associated with its properties to suppress the formation of ROS in the xanthine oxidase reaction of mitochondria and by increasing the activity of cytosolic Cu, Zn-SOD, and mitochondrial Mn-SOD, as well as catalase [51,52]. It has also been shown that “antioxidant” concentrations of Carvedilol and its metabolite BM-910228 do not affect mitochondrial respiration parameters [50]. Some studies have shown that Carvedilol, due to its uncoupling effect, can also exhibit prooxidant properties [25].

Our results, which show that Metoprolol does not have a reliable effect on the energy metabolism of the myocardium, coincide with the data of other researchers. Thus, it was shown that this selective beta-blocker, prescribed for the treatment of CHF [12], does not improve the mitochondrial ultrastructure after the introduction of doxorubicin, did not reduce peroxidation processes, and did not reduce the degree of Ca^++^ overload of mitochondria [53,54,55]. Metoprolol attenuates post-infarction structural remodeling without concomitant improvement in myocardial energy metabolism in rats with chronic CHF [56]. Bisoprolol was the first beta-blocker to show clinical efficacy in heart failure [57]. Our results demonstrated the absence of a clear reliable effect of Bisoprolol on myocardial energy metabolism in CHF (except for the effect on LDH), which is consistent with other research [58]. It has been shown that the protective effect of Bisoprolol on the heart is not associated with the optimization of energy metabolism, but has other mechanisms [59]. Bisoprolol has also been shown to inhibit mitochondrial respiration and ATP synthesis in cancer cells [60]. It has been shown that the blockade of cardiac β1 receptors with traditional beta-blockers via the PKA/cAMP signaling pathway suppresses the nuclear-encoded mitochondrial protein IF1 and inhibits oxidative phosphorylation in cardiac mitochondria [60]. Nebivalol is a latest-generation beta-blocker with additional metabolitotropic properties—NO-mimetic and antioxidant—and is actively used in the treatment of arterial hypertension and CHF [61]. To date, there are no complete data on the effect of Nebivalol on myocardial energy metabolism in CHF, and it is difficult for us to compare our modest results with the data of other researchers. It is known that Nebivalol is an ROS scavenger and is able to protect mitochondrial membranes, affect various mechanisms of mitoptosis, increase ATP, creatine phosphate, and normalize the [lactate]/[pyruvate] ratio. Moreover, the effect on energy metabolism is not associated with its NO-mimetic effect [62,63]. There are, however, other studies demonstrating the involvement of Nebivalol in enhancing the formation of mitochondrial dysfunction in cancer cells [60,64]. Other researchers do not confirm a direct negative effect of Nebivalol on mitochondria in non-tumor cells, which emphasizes its specificity and excludes any antimitotic toxicity [60,65].

## 5. Conclusions

Thus, the conducted studies have established an ambiguous effect of beta-blockers on myocardial energy metabolism in the doxorubicin model of CHF. Traditional second-generation beta-blockers (Metoprolol and Bisoprolol) did not affect the studied indices of energy metabolism, and third-generation beta-blockers with additional properties—Carvedilol and, especially, Nebivalol and Hypertril—improved myocardial energy metabolism. The use of these drugs and, especially, Hypertril in CHF led to an increase in the content of ATP and ADP against the background of a decrease in AMP and lactate with a simultaneous increase in the activity of SDH, NAD-MDH, and malate concentration in various fractions of the myocardial homogenate of rats with CHF. Beta-blockers with additional properties and, especially, Hypertril, reduced the opening of the mitochondrial pore (MP) and increased the charge of the inner membrane of the myocardial mitochondria. The revealed facts indicate the possible influence of Nebivalol and Hypertril on individual links of myocardial energy metabolism (activation of compensatory energy shunts under ischemic conditions), as well as a possible mitoprotective effect. The results obtained will help to expand our understanding of the effect of beta-blockers of various generations used to treat cardiovascular diseases on energy metabolism, and are also an experimental justification for the practical choice of these drugs in the complex therapy of CHF.

## Figures and Tables

**Figure 1 biomedicines-12-01957-f001:**
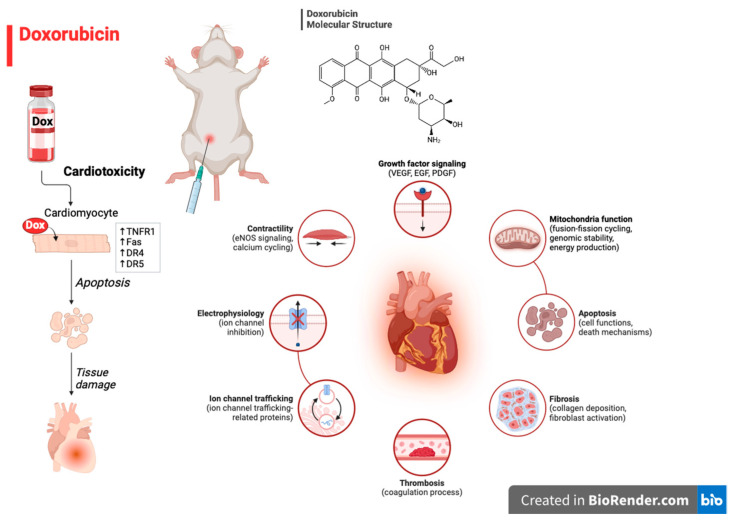
Doxorubicin-induced cardiotoxicity mechanisms.

**Table 1 biomedicines-12-01957-t001:** The influence of Hypertril and the reference drug on the content of adenyl nucleotides in the cytosolic fraction of the myocardium of animals with experimental CHF.

Group of Animals	ATP, µmol/g Tissue	ADP, µmol/g Tissue	AMP, µmol/g Tissue
Intact (*n* = 10)	3.8 ± 0.21	0.57 ± 0.02	0.154 ± 0.007
CHF (control) (*n* = 6)	1.9 ± 0.12 ^1^	0.33 ± 0.017 ^1^	0.27 ± 0.014 ^1^
CHF + Hypertril, 3.5 mg/kg (*n* = 19)	2.7 ± 0.12 *^1^	0.41 ± 0.010 *^1^	0.169 ± 0.011 *^1^
CHF + Carvedilol 50 mg/kg (*n* = 10)	2.2 ± 0.09 ^1^	0.36 ± 0.017 ^1^	0.25 ± 0.017 ^1^
CHF + Nebivalol, 10 mg/kg (*n* = 16)	2.3 ± 0.07 *^1^	0.36 ± 0.021 ^1^	0.23 ± 0.011 *^1^
CHF + Bisoprolol, 10 mg/kg (*n* = 16)	1.85 ± 0.14 ^1^	0.35 ± 0.033 ^1^	0.26 ± 0.034 ^1^
CHF + Metoprolol, 15 mg/kg	1.9 ± 0.11 ^1^	0.34 ± 0.054 ^1^	0.25 ± 0.017 ^1^

In parentheses is the number of animals that survived at the end of the experiment. *—changes are significant in relation to animals in the control group (*p* < 0.05); ^1^—changes are significant in relation to animals of the intact group (*p* < 0.05).

**Table 2 biomedicines-12-01957-t002:** The effect of Hypertril and reference drug on the content of energy metabolism intermediates in the cytosol of the heart of animals with experimental CHF.

Group of Animals	Lactate, µmol/g Tissue	Malate, µmol/g Tissue	Pyruvate, µmol/g Tissue
Intact (*n* = 10)	5.2 ± 0.31	0.36 ± 0.02	0.15 ± 0.01
CHF (control) (*n* = 6)	12.1 ± 0.87 ^1^	0.11 ± 0.01 ^1^	0.071 ± 0.001 ^1^
CHF + Hypertril, 3.5 mg/kg (*n* = 19)	7.4 ± 0.44 *^1^	0.21 ± 0.01 *^1^	0.088 ± 0.001 *^1^
CHF + Carvedilol 50 mg/kg (*n* = 10)	10.2 ± 0.92 ^1^	0.12 ± 0.02	0.072 ± 0.001 ^1^
CHF + Nebivalol, 10 mg/kg (*n* = 16)	8.10 ± 0.54 *^1^	0.14 ± 0.01 *^1^	0.080 ± 0.001 *^1^
CHF + Bisoprolol, 10 mg/kg(*n* = 16)	11.5 ± 0.42 ^1^	0.12 ± 0.01 ^1^	0.072 ± 0.003 ^1^
CHF + Metoprolol, 15 mg/kg	11.1 ± 1.2	0.12 ± 0.01 ^1^	0.065 ± 0.002

In parentheses is the number of animals that survived at the end of the experiment. *—changes are significant in relation to animals in the control group (*p* < 0.05); ^1^—changes are significant in relation to animals of the intact group (*p* < 0.05).

**Table 3 biomedicines-12-01957-t003:** The effect of Hypertril and reference drug on energy metabolism indices in the mitochondrial fraction of cardiac tissue of animals with experimental CHF.

Group of Animals	Lactate, µmol/g Tissue	NAD-MDH, μmol/mg Protein/min	SDH, nmol/mg Protein/min	MPTP Opening, ∆E 540 nm	MMP, (Ψ)	ATP, µmol/g Tissue
Intact (*n* = 10)	1.5 ± 0.08	1.71 ± 0.08	5.5 ± 0.28	0.052 ± 0.002	52.1 ± 3.2	2.7 ± 0.12
CHF (control) (*n* = 6)	2.7 ± 0.14 ^1^	0.65 ± 0.04 ^1^	1.8 ± 0.10 ^1^	0.63 ± 0.022 ^1^	18.2 ± 1.0 ^1^	1.2 ± 0.03 ^1^
CHF + Hypertril, 3.5 mg/kg (*n* = 19)	1.7 ± 0.15 *	1.14 ± 0.08 *^1^	3.1 ± 0.17 *^1^	0.31 ± 0.005 *^1^	33.2 ± 2.4 *^1^	1.60 ± 0.15 *^1^
CHF + Carvedilol 50 mg/kg (*n* = 10)	2.5 ± 0.19 ^1^	0.67 ± 0.01 ^1^	2.2 ± 0.14 *^1^	0.44 ± 0.011 *^1^	25.4 ± 1.2 *^1^	1.3 ± 0.11 ^1^
CHF + Nebivalol, 10 mg/kg (*n* = 16)	2.2 ± 0.10 *^1^	0.79 ± 0.04 *^1^	2.6 ± 0.11 *^1^	0.47 ± 0.007 *^1^	25.7 ± 1.8 *^1^	1.42 ± 0.07 *^1^
CHF + Bisoprolol, 10 mg/kg (*n* = 16)	2.3 ± 0.18 *^1^	0.67 ± 0.02 ^1^	2.0 ± 0.22 ^1^	0.63 ± 0.015 ^1^	18.8 ± 2.7 ^1^	1.2 ± 0.25 ^1^
CHF + Metoprolol, 15 mg/kg	2.5 ± 0.21 ^1^	0.68 ± 0.03 ^1^	2.1 ± 0.19 ^1^	0.65 ± 0.011 ^1^	19.3 ± 3.0 ^1^	1.2 ± 0.10 ^1^

In parentheses is the number of animals that survived at the end of the experiment. *—changes are significant in relation to animals in the control group (*p* < 0.05); ^1^—changes are significant in relation to animals of the intact group (*p* < 0.05). NAD-MDH—Nicotinamide adenine dinucleotide-malate dehydrogenase. SDH—succinate dehydrogenase. MPTP—mitochondrial permeability transition pore. MMP—mitochondrial membrane potential.

## Data Availability

The original contributions presented in the study are included in the article, further inquiries can be directed to the corresponding authors.

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
