# Peer review of "Beta-Blockers of Different Generations: Features of Influence on the Disturbances of Myocardial Energy Metabolism in Doxorubicin-Induced Chronic Heart Failure in Rats"

_biomedicines, 2024, doi:10.3390/biomedicines12091957_

Round 1

Reviewer 1 Report

Comments and Suggestions for Authors

The authors used a doxorubicin model to induce chronic heart failure in rats. Subsequently, they compared several beta blockers and their new drug Hypertril with respect to myocardial energy metabolism. The main results are that traditional (second generation) selective beta-1-blockers had no effect on the energy metabolism, while the third-generation beta-blockers and especially Hypertril improved the tested parameters of energy metabolism.

The methods are clearly described, and the results support the conclusions. Nevertheless, I have some points to be addressed:

1)      It is not clear to me that Hypertril is really a beta-blocker. The previous study, which was published in the European Journal of Pharmacology, also only demonstrated the blood pressure lowering effect in a rat model of hypertension (SHR), but not the underlying mechanism. Therefore, the wording should be toned down throughout the manuscript: just talk about the new anti-hypertensive drug Hypertril. Or provide data to clearly characterize it as a beta-blocker.

2)      I am wondering that only biochemical data are presented. Please also provide functional data on the doxorubicin model to demonstrate the successful induction of heart failure.

3)      The limitations of the study must also be mentioned. Only heart failure caused by doxorubicin was examined here. For example, it is known that the cardiotoxic effects of doxorubicin are due to oxidative stress, mitochondrial dysfunction and impaired energy metabolism. However, chronic heart failure can also have many other causes in patients. Therefore, this is a limitation. In addition, an improvement in cardiac function was not shown.

Author Response

We thank the reviewer for the evaluation and feedback. 

Q1. 1)      It is not clear to me that Hypertril is really a beta-blocker. The previous study, which was published in the European Journal of Pharmacology, also only demonstrated the blood pressure lowering effect in a rat model of hypertension (SHR), but not the underlying mechanism. Therefore, the wording should be toned down throughout the manuscript: just talk about the new anti-hypertensive drug Hypertril. Or provide data to clearly characterize it as a beta-blocker.

Answer1. The Hypertril molecule was synthesized and then underwent a fairly long stage of preliminary experimental studies. Thus, data were obtained that Hypertril competes for beta 1 receptors of the isolated heart when loaded with beta-adrenomimetics. These results were also confirmed under conditions of the whole organism when loaded with isadrine. These studies were carried out in accordance with the requirements of the State Expert Center of the Ministry of Health of Ukraine, which are consistent with similar European recommendations (see references below, [1-3]).

Q2. 2)      I am wondering that only biochemical data are presented. Please also provide functional data on the doxorubicin model to demonstrate the successful induction of heart failure.

Answer2. Electrocardiographic studies also confirmed the beta-blocking effect of Hypertril on the doxorubicin CHF model. A previous study also presented functional data demonstrating successful induction of heart failure [4]. Moreover, we described the morpho-functional changes in the myocardium of rats in this model of CHF induced by doxorubicin [5].

Q3. 3)      The limitations of the study must also be mentioned. Only heart failure caused by doxorubicin was examined here. For example, it is known that the cardiotoxic effects of doxorubicin are due to oxidative stress, mitochondrial dysfunction and impaired energy metabolism. However, chronic heart failure can also have many other causes in patients. Therefore, this is a limitation. In addition, an improvement in cardiac function was not shown.

Answer3. Indeed, it was previously demonstrated that doxorubicin causes activation of oxidative stress, damage to mitochondria and the formation of mitochondrial dysfunction, initiation of apoptosis of cardiomyocytes, and subsequently, after the cessation of doxorubicin administration, persistent, irreversible changes in the structure of the myocardium occur, leading to its disorders characteristic of CHF (references below, [6-8]).

References:

  1. Report “Preclinical study of specific biological activity (antianginal, antihypertensive) of the drug “Gippertril” with parenteral administration” / edited by Yu.M. Kolesnik. - Zaporozhye, 2012. - 276 p. (Очет «Доклиническое исследование специфической биологической активности (антиангинальная, антигипертензивная) препарата «Гипертрил» при парентеральном введении» / под ред Ю.М.Колесника.- Запорожье,2012.-276с.);
  2. Study of β - adrenergic blocking activity in the series of 1-alkyl (carboxyalkyl)-4-ylidenamino-1,2,4-triazolium derivatives / I. F. Belenichev, Yu. A. Volchik, L. I. Kucherenko, E. A Nagornaya, N.V. Parnyuk, I.A. Mazur, N.A. Avramenko, E.A. Portnaya // Experimental and clinical physiology and biochemistry. – 2014. - No. 3. – P. 24-32 (Исследование β - адреноблокирующей активности в ряду производных 1-алкил (карбоксиалкил)-4-илиденамино-1,2,4-триазолия / И. Ф. Беленичев, Ю. А. Волчик, Л. И. Кучеренко, Е. А. Нагорная, Н. В. Парнюк, И. А. Мазур, Н. А. Авраменко,  Е. А. Портная // Експериментальна та клінічна фізіологія і біохімія. – 2014. - № 3. – С. 24-32);
  3. Tishcin V.S. Clinical and experimental research of efficiency of means of metabolic correction in the combined therapy of acute heart attack .-Disertation - Zaporozhye, 1989. – p.402
  4. Goncharov O, Belenichev I, Abramov A, Popazova O, Kucherenko L, Bukhtiyarova N, Pavliuk I (2023) Influence of experimental heart failure therapy with different generations of β-adrenergic blockers on Cardiac Electrical Activity (ECG) and Autonomic Regulation of Heart Rhythm (ARHR). Pharmacia 70(4): 1157–1165. https://doi.org/10.3897/pharmacia.70.e110924
  5. Bak PG, Belenichev IF, Kucherenko LI, Abramov AV, Khromylоva OV (2021) Morpho-functional indicators changes of rats’ myocardium in experimental doxorubicin-induced chronic heart failure and its pharmacological modulation with new 4-amino-1,2,4-triazole derivative. Pharmacia 68(4): 919–925. https://doi.org/10.3897/pharmacia.68.e75298
  6. Belenichev I, Bak P, Popazova O, Ryzhenko V, Bukhtiyarova N, Puzyrenko A. INTEGRATIVE AND BIOCHEMICAL PARAMETERS IN RATS IN THE SIMULATION OF DOXORUBICIN CHRONIC HEART FAILURE AND DURING THE USE OF Β-ADRENERGIC BLOCKERS. J. Fac. Pharm. Ankara. January 2023;47(1):228-238. doi:10.33483/jfpau.1131302;
  7. Khloponin, D.P. (2009). Analysis of possible mechanisms of pharmacological reversal of cardiac remodeling in chronic heart failure (Doctoral dissertation). Volgograd State Medical University, Volgograd.;
  8. Podyacheva EY, Kushnareva EA, Karpov AA, Toropova YG. Analysis of Models of Doxorubicin-Induced Cardiomyopathy in Rats and Mice. A Modern View From the Perspective of the Pathophysiologist and the Clinician. Front Pharmacol. 2021 Jun 3;12:670479. doi: 10.3389/fphar.2021.670479.

Reviewer 2 Report

Comments and Suggestions for Authors

The manuscript by Belenichev et al. aims to describe the effects of second- and third-generation beta-blockers on myocardial energy metabolism in rats with doxorubicin-induced heart failure. Overall, the presentation of the manuscript could be improved. There are numerous typographical errors, including incorrect use of the degree symbol and improper word usage, as well as several verbose sentences that require restructuring. Long sentences should be divided into shorter, clearer sentences. Abbreviations should be defined upon their first appearance. The manuscript would benefit from proofreading by a native speaker. Specific comments for enhancement are provided below.

1. The title should be rephrased for conciseness and should include the specific model used (e.g., rat, rabbit).

2. Abstract: The phrase "cardiovascular diseases on energy metabolism" is unclear.

3. Introduction:

It is quite lengthy. It should include brief introduction on myocardial energy metabolism (malate/lactate/pyruvate).

Lines 43-44 are unclear.

Line 52: What is the importance of the sentence?

Line 80: What is macroerg? What is the difference between ROS production and oxidative stress?

Lines 83-93: Unclear, need to be rephrased.

The last sentence of this section should be revised into a complete sentence.

4. Materials and Methods:

What is the difference between quarantine and acclimatization? They are distinct processes.

The description of the experimental model reads more like a summary of the findings rather than a procedural description.

The sources of drugs and chemicals should be described earlier in the methodology section. For the apparatus used, the brand names and countries of origin should also be provided.

Lines 139-142: A repetition.

Lines 144-145: An incomplete sentence.

Sections 2.6.1–2.6.7 need to be revised to use passive voice.

5. Results

It is recommended to separate the Results and Discussion sections. In the current version, they are not well organized.

Does 'intact' refer to the normal control group? It would be clearer to rename it as 'control,' and rename the CHF (control) group as 'CHF.'

Line 327: Remove "We".

Table 3: What is IP?

All abbreviations used in the tables should be defined in the legends.

6. Discussion

Lines 358-369: What is the relevance of this paragraph to the current study?

Overall, this section contains numerous descriptions of findings. The Discussion should focus solely on interpreting the findings and should not include group names, p-values, or references to tables. It should provide explanations for the observed changes and discuss why the effects of hypertril, carvedilol, and nebivolol differ from those of classical beta-blockers. Are the antioxidant capacity and NO-mimetic effects of hypertril and carvedilol significantly higher than those of metoprolol? Could this be attributed to structural differences?

7. Conclusion

It reads like another discussion and needs to be revised for conciseness.

Comments on the Quality of English Language

It requires proofreading by a native speaker.

Author Response

We thank the Reviewer for the evaluation and detailed feedback. The responses are included below. 

Q1. The title should be rephrased for conciseness and should include the specific model used (e.g., rat, rabbit).

Answer1. We have revised the title.

Q2. Abstract: The phrase "cardiovascular diseases on energy metabolism" is unclear.

Answer2. We have revised this phrase (and some other phrases) to improve the clarity.

Q3. Introduction:

It is quite lengthy. It should include brief introduction on myocardial energy metabolism (malate/lactate/pyruvate).

Lines 43-44 are unclear.

Line 52: What is the importance of the sentence?

Line 80: What is macroerg? What is the difference between ROS production and oxidative stress?

Lines 83-93: Unclear, need to be rephrased.

The last sentence of this section should be revised into a complete sentence.

Answer 3. We have now revised the text based on the Reviewer's suggestions

Q4. Materials and Methods:

What is the difference between quarantine and acclimatization? They are distinct processes.

The description of the experimental model reads more like a summary of the findings rather than a procedural description.

The sources of drugs and chemicals should be described earlier in the methodology section. For the apparatus used, the brand names and countries of origin should also be provided.

Lines 139-142: A repetition.

Lines 144-145: An incomplete sentence.

Sections 2.6.1–2.6.7 need to be revised to use passive voice.

Answer4. We have now revised the text based on the Reviewer's suggestions

Q5. Results

It is recommended to separate the Results and Discussion sections. In the current version, they are not well organized.

Does 'intact' refer to the normal control group? It would be clearer to rename it as 'control,' and rename the CHF (control) group as 'CHF.'

Line 327: Remove "We".

Table 3: What is IP?

All abbreviations used in the tables should be defined in the legends.

Answer 5. We have now revised the text based on the Reviewer's suggestions

Q6. Discussion

Lines 358-369: What is the relevance of this paragraph to the current study?

Overall, this section contains numerous descriptions of findings. The Discussion should focus solely on interpreting the findings and should not include group names, p-values, or references to tables. It should provide explanations for the observed changes and discuss why the effects of hypertril, carvedilol, and nebivolol differ from those of classical beta-blockers. Are the antioxidant capacity and NO-mimetic effects of hypertril and carvedilol significantly higher than those of metoprolol? Could this be attributed to structural differences?

Answer6. We have now revised the text based on the Reviewer's suggestions

Q7. Conclusion

It reads like another discussion and needs to be revised for conciseness.

Answer7. We have now revised the entire manuscript's text based on the Reviewer's suggestions.

Round 2

Reviewer 1 Report

Comments and Suggestions for Authors

The authors have answered all objections to my satisfaction.